# Prior Appendicectomy and Gut Microbiota Re-Establishment in Adults after Bowel Preparation and Colonoscopy

**DOI:** 10.3390/biomedicines12091938

**Published:** 2024-08-23

**Authors:** Amelia J. McGuinness, Martin O’Hely, Douglas Stupart, David Watters, Samantha L. Dawson, Christopher Hair, Michael Berk, Mohammadreza Mohebbi, Amy Loughman, Glenn Guest, Felice N. Jacka

**Affiliations:** 1The Institute for Mental and Physical Health and Clinical Translation (IMPACT), School of Medicine and Barwon Health, Deakin University, Geelong, VIC 3220, Australia; 2Murdoch Children’s Research Institute, Parkville, VIC 3052, Australia; 3School of Medicine, Deakin University, Geelong, VIC 3220, Australia; 4Department of Surgery, University Hospital Geelong, Barwon Health, Geelong, VIC 3220, Australia; 5Department of Gastroenterology, Epworth Hospital, Waurn Ponds, VIC 3216, Australia; 6Orygen, The National Centre of Excellence in Youth Mental Health, Centre for Youth Mental Health, Florey Institute for Neuroscience and Mental Health and the Department of Psychiatry, The University of Melbourne, Melbourne, VIC 3052, Australia; 7Biostatistics Unit, Faculty of Health, Deakin University, Burwood, VIC 3125, Australia; 8Centre for Adolescent Health, Murdoch Children’s Research Institute, Melbourne, VIC 3052, Australia; 9College of Public Health, Medical & Veterinary Sciences, James Cook University, Townsville, QLD 4814, Australia

**Keywords:** bowel preparation, gut microbiota, colonoscopy, appendix, appendicectomy, appendectomy

## Abstract

Emerging evidence suggests that the human vermiform appendix is not a vestigial organ but rather an immunological organ of biological relevance. It is hypothesised that the appendix acts as a bacterial ‘safe house’ for commensal gut bacteria and facilitates re-inoculation of the colon after disruption through the release of biofilms. To date, no studies have attempted to explore this potential mechanistic function of the appendix. We conducted a pre-post intervention study in adults (n = 59) exploring re-establishment of the gut microbiota in those with and without an appendix after colonic disruption via bowel preparation and colonoscopy. Gut microbiota composition was measured one week before and one month after bowel preparation and colonoscopy using 16S rRNA sequencing. We observed between group differences in gut microbiota composition between those with (n = 45) and without (n = 13) an appendix at baseline. These differences were no longer evident one-month post-procedure, suggesting that this procedure may have ‘reset’ any potential appendix-related differences between groups. Both groups experienced reductions in gut microbiota richness and shifts in beta diversity post-procedure, with greater changes in those without an appendix, and there were five bacterial genera whose re-establishment post-procedure appeared to be moderated by appendicectomy status. This small experimental study provides preliminary evidence of a potential differential re-establishment of the gut microbiota after disruption in those with and without an appendix, warranting further investigation into the potential role of the appendix as a microbial safe house.

## 1. Introduction

The role of the human vermiform appendix is contentious. Long thought to be a vestigial organ, nascent research now suggests that the appendix is more than just an evolutionary remnant of the caecum. Rather, it is an active immunological organ made up of gut-associated lymphoid tissue and immune cells with a potential luminal reservoir [1,2,3,4,5]. Further, the appearance of the appendix early in gestation [6], the rarity of appendiceal malformation or absence (agenesis) [7], and the continuous presence of the appendix in humans (and appendix-like structures in other mammals) across millennia [1] contradicts evolutionary redundancy and suggests ongoing biological relevance.

There is growing evidence to support the concept of the colonic gut microbiota being essential for host health, with disruptions to gut ecosystem homeostasis being commonly associated with diseases including colorectal cancer [8], obesity [9], and mental disorders [10]. A study of extracted appendices revealed a rich appendiceal microbiota, with greater diversity than that of the colonic microbiota, including bacteria only previously found in oral cavities [11]. Thus, it has been postulated that a function of the appendix may be to act as a microbial reservoir and to protect important gut bacteria during gastrointestinal disruption, such as infection or diarrhoeal disease [4,12]. Indeed, the positioning of the appendix at the ileo-caecal junction leaves it relatively protected from the faecal stream [6,12], further supporting the potential for it to act as a microbial ‘safe house’ that protects commensal gut bacteria.

In addition to this potential structural protective function, the appendix supports the production and maintenance of biofilms [12]. Biofilms are abundant in the appendix and caecum, then decrease in density along the length of the large intestine, with limited to no presence in the distal colon and rectum [12]. Biofilms comprise an extracellular matrix of host and microbial components that line portions of the gastrointestinal tract and assist with defence against pathogenic invasion [12]. These biofilms undergo constant cycles of shedding and regeneration and are believed to release bacteria from their matrix into the intestinal lumen when required [2,12,13]. Thus, it is postulated that the appendix may facilitate reinoculation of the gut microbiota after disruption through the release of biofilms into the colon, which can then adhere to the luminal surfaces and shed bacteria to assist with homeostatic regulation and colonisation resistance [4,12].

Thus, the aim of this study was to examine the possible influence of prior appendicectomy on gut microbiota composition and re-establishment after induced disruption via bowel preparation and colonoscopy (hereafter referred to as the ‘procedure’). We firstly aimed to explore whether people without an appendix (i.e., with prior appendicectomy) had a different gut microbiota composition at baseline compared to those with an appendix. Secondly, we aimed to examine whether the absence of an appendix influenced the trajectory of gut microbiota re-establishment post-procedure.

## 2. Materials and Methods

### 2.1. Study Design

The data presented in this manuscript originate from the Micro-Scope study, a pre–post intervention study designed to investigate changes in depressive symptoms and gut microbiota composition following bowel preparation and colonoscopy. The findings are reported in accordance with established guidelines, including the Strengthening the Reporting of Observational Studies in Epidemiology Statement for observational studies [14] and the Strengthening the Organising and Reporting of Microbiome Studies checklist [15]. Ethical approval for this study was obtained from the Barwon Health (#15-129), Epworth Healthcare (#EH2016-146), and Deakin University (#2016-391) human research ethics committees. Microbial data were not shared in an online public repository due to the absence of consent for data sharing. This study was pre-registered on the Open Science Framework: https://osf.io/j69tk (accessed on 18 August 2024).

### 2.2. Study Participants and Setting

The study included community-dwelling adults referred for colonoscopy between May 2017 and November 2018 at University Hospital Geelong, Australia. Participants were recruited during their initial outpatient consultation with the General Surgery or Gastroenterology services. Any adults referred for colonoscopy during the study period were eligible for inclusion unless they were heavily reliant on medical care or unable to provide informed consent (e.g., due to language barriers, significant intellectual or cognitive disability). No exclusions were made based on recent antibiotic usage. Eligible participants were referred to a research team member for detailed study discussions and consent. Colonoscopies were conducted at University Hospital Geelong or Epworth Hospital Geelong. Participants diagnosed with cancer post-colonoscopy were withdrawn, and their follow-up data were not included.

### 2.3. Data Collection

Baseline data were obtained one week before the procedure. Participants completed paper-based questionnaires and collected a fresh faecal sample in a sterile collection jar with a scoop lid at home. The sample was stored in their home freezer (−20 °C) for approximately one week until it was transported on ice to a research team member on the day of their colonoscopy. Upon receipt, it was transferred to a −80 °C freezer for storage until DNA extraction. One month after the procedure, participants completed another set of questionnaires and provided a final faecal sample at home as above.

### 2.4. Intervention

Bowel preparation followed standard procedures prescribed by the colonoscopy service. All participants were advised to initiate a low-fibre (‘white’) diet two days before their colonoscopy, followed by a 12–24 h period of clear fluids before the procedure. They were instructed to consume a sodium picosulfate-based bowel preparation product in three separate doses. The adequacy of bowel preparation was assessed during the colonoscopy by the endoscopist using a modified overall Boston Bowel Preparation Scale score [16].

### 2.5. Outcome Measures

The primary outcome of this study was the composition of the gut bacterial microbiota. Alpha-diversity of the gut bacterial microbiota was assessed using the Shannon index—a composite measure of richness and evenness within a sample—as well as the count of taxa present at the genus level. Beta-diversity was quantified using the Aitchison distance metric, and differential taxonomic abundances at the genus level were evaluated using centred log-ratio (CLR) transformed count abundance data.

### 2.6. Covariates

Age (in years) at the time of recruitment and sex (male/female/other) were extracted from medical records. A triage nurse recorded the participants’ heights and weights during their initial outpatient consultation, facilitating the computation of a body mass index (kilograms/meter^2^). Participants self-reported their residential postcode and suburb, allowing for the determination of socioeconomic status via an area-based metric called the Index of Relative Socioeconomic Advantage and Disadvantage (IRSAD) [17]. Each suburb has an IRSAD classification on a scale of 1–10, where a lower score suggests greater disadvantage. Self-reported information included smoking status, lifetime medical conditions (including depression), and current medication usage. Diet quality was assessed using the Simple Dietary Questionnaire [18], with the total score (out of 100) indicating adherence to the Australian Dietary Guidelines; higher scores denoted greater adherence [19]. The ROME III Diagnostic Questionnaire for Adult Functional Gastrointestinal Disorders was used to establish whether participants met the criteria for irritable bowel syndrome at baseline [20]. Depressive symptoms were assessed using the depression sub-score of the Hospital Anxiety and Depression Scale [21]; the severity of depressive symptoms was measured using the Patient Health Questionnaire-9 [22]; anxiety symptoms were evaluated using the anxiety sub-score of the Hospital Anxiety and Depression Scale [21]; overall quality of life, as well as psychosocial and physical quality of life, were measured using the Assessment of Quality of Life-8 [23]; and stress levels were quantified using the Perceived Stress Scale [24]. Details on colonoscopy indications and outcomes were extracted from medical records.

### 2.7. DNA Extraction

Microbial DNA extraction from stool samples was performed using the commercial QIAamp Fast DNA Stool Mini Kit (QIAGEN, Hilden, Germany) according to manufacturer instructions, with an additional mechanical lysis step using PowerBead tubes (QIAGEN, Germany). Extracted DNA was stored at −80 °C until couriered on dry ice to the Australian Genomic Research Facility for sequencing.

### 2.8. Sequencing and Annotation

Sequencing of the 16S rRNA gene sequence was performed using the Illumina MiSeq (Illumina, San Diego, CA, USA) platform. The V1–V3 hypervariable region of the 16S rRNA gene was amplified by polymerase chain reaction using 27F (AGAGTTTGATCMTGGCTCAG) and 519R (GWATTACCGCGGCKGCTG) primers with a read length of 300 base pairs. Diversity profiling analysis was performed with QIIME 2 2019.7 [25]. The demultiplexed raw reads were primer trimmed and quality filtered as per Australian Genomic Research Facility protocols using the cutadapt plugin followed by denoising with DADA2 (via q2-dada2) [26]. Taxonomy was assigned to amplicon sequence variants (ASVs) using the q2-feature-classifier [27], the classify-sklearn naïve Bayes taxonomy classifier. Taxonomy was assigned using the SILVA (v.132) database.

### 2.9. Pre-Processing

Pre-processing and filtering was performed in line with recommendations from Callahan et al. [28]. Zero count bacterial features and non-bacterial taxa were removed prior to calculating the Shannon index and observed genera alpha diversity metrics. Additional filtering removed low prevalence taxa (those present in less than 5% of samples), and data were centred log-ratio transformed to calculate Aitchison distances (i.e., beta-diversity) and for differential abundance testing.

### 2.10. Statistical Analyses

Statistical analyses were performed within the RStudio [29] 4.3.1 environment. Packages used for analysis are listed within the Appendix A. Only participants with baseline and follow-up data (i.e., complete cases) were included in analyses.

We aimed to explore whether people without an appendix (i.e., with prior appendicectomy) had a different gut microbiota composition at baseline or follow-up compared to those with an appendix (i.e., between-group differences). We also aimed to examine whether the absence of an appendix influenced gut microbiota re-establishment post-procedure (i.e., within-group changes). Both between-group differences and within-group changes in alpha-diversity were calculated via generalised estimating equations assuming a Gaussian distribution with an AR(1) correlation structure to account for within participant autocorrelation. Robust standard error estimates are reported for all GEE models. Between-group differences and within-group changes in beta-diversity were calculated using permutational analysis of variance (adonis2) with 999 permutations, stratified by participant ID when appropriate considering the paired nature of the data. Differential abundance analyses at the genus level were calculated using general linear mixed models using the Maaslin2 package with the minimum abundance and prevalence set to zero and a CLR transformation. For between-group differences in genera at baseline and follow up, an appendicectomy group was included as the fixed effect in the model. For estimating within-group changes in genera pre-post procedure, the time point was the fixed effect and the participant identifier was the random effect. The Benjamini–Hochberg procedure was applied to control the false discovery rate, with associations yielding an adjusted *p*-value (i.e., *q*-value) under 0.05 reported in results.

We used CLR transformation to analyse bacterial abundances instead of relative abundances due to the compositional nature of the microbiome data. Relative abundances can be misleading because they represent parts of a whole and depend on the total count of all taxa, which can vary between samples. The CLR transformation addresses this issue by converting the data into log-ratios (after adding a pseudo-count of 1 to avoid taking a log of zero), allowing for more accurate and balanced comparisons. This method ensures that each taxon’s abundance is expressed relative to the geometric mean of all taxa in a sample, preserving the relationships between taxa and making the data more suitable for statistical analysis.

We also conducted additional post hoc exploratory analyses to explore whether prior appendicectomy influenced the re-establishment of any bacteria at the genus level. These analyses were calculated using the Maaslin2 package with minimum abundance and the prevalence set to zero, a centred log-ratio transformation, the participant identifier as the random effect, and appendicectomy status, time point, and an appendicectomy status by time point interaction as fixed effects. As this was an exploratory analysis, we did not control the false discovery rate, with taxa below a *p*-value of 0.05 being reported in the results.

Given the small sample size in our study, it is important to interpret the results cautiously. Statistical significance should be considered in conjunction with effect sizes and the context of the study.

## 3. Results

### 3.1. Participant Characteristics

A total of 136 participants provided informed consent at the time of their initial outpatient appointment, 86 of which were successfully contacted prior to colonoscopy and provided baseline data. Twenty participants were lost to follow-up, five participants were excluded from analyses due to inadequate bowel preparation (rated as ‘poor’ by their endoscopist), two were excluded due to cancer diagnosis post-procedure, and one was excluded due to uncertainty regarding appendicectomy status. Therefore, a total of 58 participants were included in the present analyses and provided both pre- and post-procedure stool samples, with 13 participants (22%) reporting a prior appendicectomy (i.e., no appendix). Baseline characteristics are presented in Table 1 and procedural outcomes are described in Appendix A.

Those without an appendix had a lower average body mass index compared to those with an appendix (*p* = 0.007) and had lower average depression symptom scores (*p* = 0.002), lower average depression severity scores (*p* = 0.005), and weak evidence of lower average anxiety scores (*p* = 0.054). Those without an appendix also reported higher average psychosocial (*p* = 0.040), physical (*p* = 0.014), and total (*p* = 0.006) quality of life scores compared to those with an appendix.

There were no statistically significant differences between groups with respect to age, sex, smoking status, meeting the diagnostic criteria for irritable bowel syndrome, or average diet quality.

### 3.2. Between-Group Differences in Gut Microbiota Composition at Baseline and Follow Up

There was no statistical evidence of a difference in alpha-diversity between groups at baseline or follow up (Appendix A). However, there was weak evidence of a difference in beta-diversity between groups at baseline (R^2^ = 0.02, *p* = 0.084) (Appendix A), and this had been attenuated at follow up (R^2^ = 0.02, *p* = 0.154) (Appendix A).

At baseline, higher average CLR-transformed abundances of an uncultured Coriobacteriales incertae sedis genus (β = 2.67, 95%CI: 1.45, 3.88, *q* = 0.008) and *Ruminococcaceae DTU089* (β = 1.77, 95%CI: 0.81, 2.73, *q* = 0.043) were observed in those without an appendix compared to those with an appendix after adjusting for multiple comparisons. At follow up, there were no between-group differences in the CLR-transformed abundances of any taxa before or after adjusting for multiple comparisons.

### 3.3. Within-Group Changes in Gut Microbiota Composition Post-Procedure

There was statistical evidence of reduced richness one-month post-procedure compared to baseline for both groups (Appendix A). The effect estimates were slightly greater in magnitude for those without an appendix (β = −155, 95%CI: −220, −89.7) than those with an appendix (β = −128, 95%CI: −166, −89.4). There was no change in either group for the Shannon index (Appendix A).

In those with an appendix, there was a shift in beta-diversity one-month post-procedure compared to baseline (R^2^ = 0.004, *p* = 0.047) (Appendix A). There was a more pronounced shift in beta-diversity in those without an appendix (R^2^ = 0.01, *p* = 0.019), However, the explained variance for both was very small.

After adjusting for multiple comparisons, differential abundance analyses identified 22 genera in those with an appendix that were differentially abundant on the CLR scale one-month post-procedure compared to baseline, including increases in the annotated genera *Cutibacterium*, *Megamonas*, *Ruminococcaceae UCG-009*, *Gordonibacter*, *Megasphaera*, *Ruminococcaceae DTU-089*, *Lachnospiraceae UCG-010*, and *Oxalobacter* (Appendix A). In those without an appendix, there were 11 genera that were differentially abundant on the CLR scale one-month post-procedure compared to baseline, including increases in the annotated genera *Ruminococcaceae CAG-352*, *Lachnospiraceae NK4B4 group*, *Megamonas*, *Tyzzerella*, *Holdemanella*, *Mogibacterium*, and *Oxalobacter* (Appendix A).

### 3.4. Post Hoc Analyses of Appendicectomy as an Effect Modifier of Genus Re-Establishment

Exploratory analyses provided evidence (*p* < 0.05) of two-way interactions between appendicectomy status and the re-establishment of five genera; however, these were not statistically significant after adjusting for multiple comparisons (Figure 1; Appendix A). This provides weak evidence that the re-establishment of these genera differed, on average, based on whether individuals had an appendix. Post-procedure, the *Ruminococcus gauvreauii* group and *Lactonifactor* increased in those without an appendix but decreased in those with an appendix. The unidentified *Burkholderiaceae* genus increased in those with an appendix, with no change in those without an appendix. Further, *Collinsella* and the *Lachnospiraceae NK4B4* group decreased in both groups post-procedure, with *Collinsella* decreasing more in those with an appendix and the *Lachnospiraceae NK4B4* group decreasing more in those without an appendix.

## 4. Discussion

### 4.1. Summary of Findings

The aim of this study was to investigate the appendix’s potential evolutionary role as a microbial safe house by exploring gut microbiota re-establishment in individuals with and without an appendix following bowel preparation and colonoscopy. Initially, there were differences in gut microbiota composition between those with and without an appendix. However, these differences were no longer apparent one-month post-procedure. Both groups experienced reductions in microbial richness and shifts in overall community composition, with stronger changes being observed in those without an appendix. Five bacterial genera showed re-establishment patterns moderated by appendicectomy status, suggesting a differential impact of bowel preparation and colonoscopy on gut microbiota based on the presence of the appendix.

Our findings provide preliminary evidence that the appendix may act as a microbial refugium that helps to re-establish the gut microbiota after perturbation. The study’s small sample size and experimental nature limit the conclusions, but the data indicate that those without an appendix might experience more significant changes in gut microbiota composition. The results also suggest that the appendix could house important gut commensals or reflect an individual’s core microbiota, which merits further investigation.

No differences in alpha diversity were observed between groups, aligning with two previous studies [30,31], although lower alpha diversity has also been reported [32]. Weak evidence of a difference in beta diversity was noted, consistent with some prior research [30,32]. One-month post-procedure, the initial differences in gut microbiota between groups were no longer apparent, suggesting a reset effect from bowel preparation and colonoscopy. The long-term implications of these findings for human physiology remain unclear.

At baseline, participants without an appendix had higher abundances of an uncultured Coriobacteriales genus and *Ruminococcaceae DTU089*. Coriobacteriales includes genera known for fermenting glucose into lactic acid, while Ruminococcaceae contains many health-associated, butyrate-producing bacteria. Previous studies have shown varying results regarding these bacteria in individuals without an appendix, highlighting the need for replication of these findings.

Unexpectedly, those without an appendix appeared healthier at baseline, with significantly lower BMIs and better mental health scores. This contrasts with previous studies showing mixed results regarding body mass index and mental health in those without an appendix [30,31]. Potential reverse causation, where healthier individuals might be more likely to undergo an appendicectomy, could explain these findings. Given the associations between gut microbiota, body weight, and mental health, further research is needed to understand the appendix’s role in these contexts.

### 4.2. Potential Clinical Implications

The potential relationships between the appendix, the gut microbiome, and human physiology are unclear. Most studies examining the influence of the appendix in humans have been observational and have investigated the association between appendicectomy status and disease states, such as ulcerative colitis [33,34,35,36,37], Crohn’s disease [33,34,38], *Clostridioides difficile* infection [39,40,41,42], colorectal cancer [43,44,45], diverticular disease [46], ischaemic heart disease [47], myocardial infarction [48], rheumatoid arthritis [49], Parkinson’s disease [50], and mood disorders [51]. Findings in studies of each disorder have been equivocal, with little consensus as to whether prior appendicectomy is likely to be protective, a risk factor, or of little relevance to disease. Additionally, years since appendicectomy and the presence of acute appendicitis appear to influence some of these associations [36,38,43,46]. One study has further examined the relationship between appendicectomy, disease outcomes, and gut microbiome composition [52]. They observed that those with a prior appendicectomy were at an increased risk of colorectal cancer and that they had both an enrichment of colorectal cancer-promoting bacterial species and depletion of health-associated commensals compared to those without a prior appendicectomy [52]. Future research is required to replicate these findings and to explore mechanisms where appendicectomy may impact the gut microbiome and, consequently, whether this is of relevance to human health and pathophysiology.

### 4.3. Challenges to Studying the Potential Function of the Human Appendix

Studying the role of the appendix experimentally in humans is challenging. The role of the appendix in re-inoculating the gut may be more relevant in developing countries, where poor sanitation and diarrheal diseases are more common [1,4,12]. In contrast, modern hygiene, better diets, clean water, and healthcare access in Western societies may render the appendix functionally redundant, complicating its study. Consequently, our results may not be generalisable to other populations or reflective of the importance of the appendix in other circumstances, such as after infection. Additionally, the appendix’s proximal location in the colon means our faecal sample results might not capture changes occurring nearer to the appendix. Future research with improved sampling methods and a longer follow-up period is needed to provide greater insight into what might be occurring more proximally to the appendix.

### 4.4. Limitations of the Present Study

This study is the first to attempt to elucidate the evolutionary and mechanistic role of reinoculation of the colon by the human appendix. However, there are limitations that need to be noted. Our study was observational in nature, and we cannot infer a causal role between appendicectomy and reinoculation of the gut microbiota. Our sample size was very small, particularly for the appendicectomy group (n = 13). Thus, our study was not adequately powered to detect moderate to small differences and further research with a larger sample size is required. The present study used 16S rRNA gene sequencing, which is subject to variability in sequencing depth and typically has only genus-level resolution. As mentioned previously, our results may not be generalisable to other populations (e.g., developing nations) or to other forms of perturbation (e.g., diarrhoeal disease). We used faecal samples to measure gut microbiota composition, and, considering the proximal location of the appendix within the colon, whether the differences we observed are related to appendicectomy or some other aspect of health is unclear.

## 5. Conclusions

In conclusion, there was some evidence to support the concept of the appendix being an evolutionarily conserved bacterial safe house, as we observed greater changes in gut microbiota composition in those without an appendix compared to those with an appendix. Those without an appendix appeared to have better mental health, a lower body mass index, and some differences in gut microbiota composition at baseline compared to those with an appendix. Bowel preparation and colonoscopy appeared to attenuate these gut microbiota differences, and the re-establishment of some bacterial genera were moderated by appendicectomy status. However, the importance of these findings, or their meaning in the context of human physiology or evolution, is unclear. Future studies that challenge the widely repeated concept that the appendix lacks any function and is evolutionarily redundant are warranted.

## Figures and Tables

**Figure 1 biomedicines-12-01938-f001:**
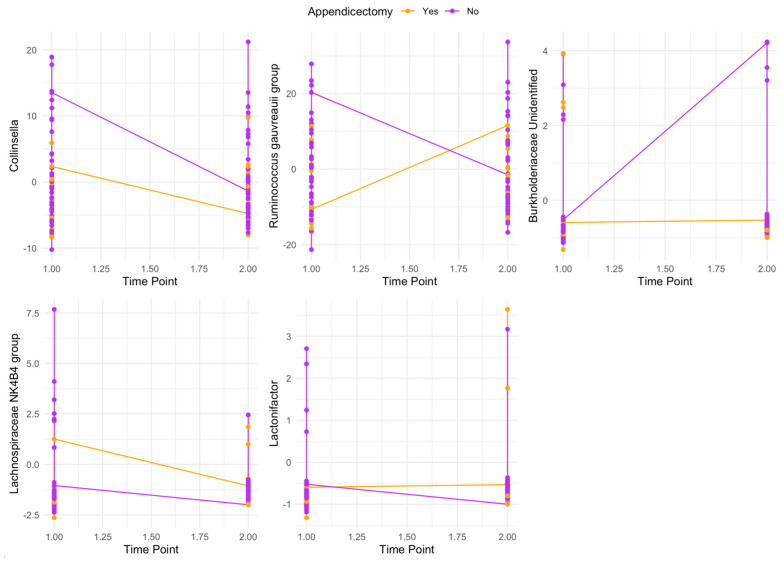
Changes in the bacterial genera with statistically significant (*p* < 0.05) two-way interactions with appendicectomy status from baseline to one-month after bowel preparation and colonoscopy. Orange, no appendix; purple, with appendix.

**Table 1 biomedicines-12-01938-t001:** Baseline characteristics of a sample of adults undergoing colonoscopy for a range of indications, stratified by those with (n = 45) or without (n = 13) an appendix.

Variable	With Appendix	Without Appendix
Participants, n (%)	45 (78%)	13 (22%)
Age, mean (SD)	58.4 (11.1)	60.5 (9.9)
Sex, female n (%)	26 (58%)	5 (39%)
BMI, mean (SD)	30.9 (6.5)	26.0 (4.8)
Currently smoking, yes n (%)	5 (11%)	1 (8%)
IBS, yes n (%)	11 (24%)	2 (15%)
Missing	1 (2.2%)	0 (0%)
Diet quality ‡, mean (SD)	44.2 (11.5)	48.9 (11.4)
Depression symptomatology, mean (SD)	4.13 (3.31)	2.15 (1.21)
Depression severity, mean (SD)	4.29 (5.88)	1.15 (2.15)
Anxiety symptomatology, mean (SD)	5.78 (3.98)	3.92 (2.56)
Physical quality of life, mean (SD)	0.689 (0.178)	0.790 (0.103)
Psychosocial quality of life, mean (SD)	0.456 (0.213)	0.581 (0.173)
Overall quality of life, mean (SD)	0.746 (0.199)	0.868 (0.106)
Gastrointestinal medications, yes n (%)		
Hyperacidity/reflux medication,	19 (42%)	3 (23%)
Digestive supplements/cholelitholytics	1 (2%)	1 (8%)
Self-reported prior gastrointestinal disease, yes n (%)		
Hiatus hernia/oesophageal reflux	5 (11%)	1 (8%)
Chronic diarrhoea	3 (7%)	0 (0%)
Chronic gastritis	1 (2%)	0 (0%)
Bowel cancer §	1 (2%)	0 (0%)
Peptic ulcer disease §	0 (0%)	1 (8%)
History of sleeve gastrectomy	1 (2%)	0 (0%)
Missing	1 (2%)	1 (8%)

‡ Diet quality was measured using a Simple Dietary Questionnaire, where the total diet quality score (out of 100) rates adherance to the Australia Dietary Guidelines, with higher scores representing greater compliance [19]. § Self-reported bowel cancer was reported by participant >15 years before study; self-reported peptic ulcer disease was reported by participant >30 years prior to study. Neither were present in the 12 months prior to study. Abbreviations: BMI, body mass index; IBS, irritable bowel syndrome.

## Data Availability

The datasets presented in this article are not readily available because consent for data sharing was not obtained from participants.

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
