# Peer review of "Prior Appendicectomy and Gut Microbiota Re-Establishment in Adults after Bowel Preparation and Colonoscopy"

_biomedicines, 2024, doi:10.3390/biomedicines12091938_

Round 1

Reviewer 1 Report

Comments and Suggestions for Authors

The authors aim to reveal the role of the human vermiform appendix in colon re-inoculation. A study (n=59) that explored gut microbiota reestablishment after colonic disruption was conducted by authors in those with and without an appendix. They claimed that differences in microbiota were observed between groups before colonoscopy, but these vanished post-procedure, indicating a potential 'reset.' Both groups experienced microbiota changes, with greater shifts in those lacking an appendix. They also found five bacterial genera that showed differential reestablishment based on appendicectomy status.

The study is interesting but no solid conclusion can be obtained from this manuscript.  

Major points: 

More convincing data are required based on the small sample size. Expanding the sample size is necessary. More efforts are needed to explain the results such as Fig.1 to help readers understand the CLR-transformed abundances.

Author Response

We would like to thank Reviewer 1 for their feedback which has ultimately improved the quality of our manuscript. Please see below for our responses to their queries:

More convincing data are required based on the small sample size. Expanding the sample size is necessary.

Thank you for this feedback. We concur that a larger sample size would enhance the robustness and generalisability of our findings. Given that our study utilised historical data collected from mid-2017 to late-2018, we are currently unable to expand our sample size for these specific analyses. However, in the limitations section we acknowledge the necessity of further research with a larger cohort and hope to explore this hypothesis in a more extensive sample.

More efforts are needed to explain the results such as Fig.1 to help readers understand the CLR-transformed abundances.

Thank you for this feedback. We agree that CLR transformation can be a complex and challenging topic to interpret. In response, we have added a detailed explanation of CLR transformation and the rationale for choosing this method over relative abundances in the Methods section. Additionally, we have expanded the description and interpretation of the results presented in Figure 1 to enhance clarity and help readers better understand the CLR-transformed abundances.

Reviewer 2 Report

Comments and Suggestions for Authors

1.       The data presented is acquired from old repositories of 2017-2008. Why did the authors report this late? Please discuss and justify.

2.       What specific bacterial genera showed a significant increase or decrease before or after appendectomy?

3.       Did the authors consider the impact of antibiotics taken during this procedure and its impact on bacterial diversity?  Please discuss and justify.

4.       Please include the figures representing alpha and beta diversity as a whole.

5.       What are the conclusions from Figure 1 while it is written in the legend as weak evidence? Please justify.

6.       There are many limitations associated with this study. Authors need to include a separate section describing the limitations of the present study.

Author Response

We would like to thank Reviewer 2 for their feedback which has ultimately improved the quality of our manuscript. Please see below for our responses to their queries:

The data presented is acquired from old repositories of 2017-2008. Why did the authors report this late? Please discuss and justify.

Thank you for your query regarding the timeline of our study. The data were collected between 2017 and the end of 2018. Following data collection, we sought funding for sequencing in 2019. However, our study was conducted in Victoria, Australia, which had some of the longest and most restrictive COVID lockdowns in the world between 2020 and 2022 (e.g. between March 2020 and October 2021, we experienced six lockdowns totalling 262 days); this severely impacted our access to laboratory facilities and our ability to finalise the study. These delays were beyond our control. We were able to resume our work in 2023, and we are now publishing our findings in 2024. We appreciate your understanding of these unprecedented circumstances.

What specific bacterial genera showed a significant increase or decrease before or after appendectomy?

Thank you for your question. Our study was observational and focused on individuals with a history of appendicectomy, rather than examining changes in bacterial genera immediately before or after the procedure. Consequently, we do not have data on specific bacterial genera that increase or decrease directly following appendicectomy. We recognise the importance of this area and are interested in pursuing future research to explore these dynamics in greater detail.

Did the authors consider the impact of antibiotics taken during this procedure and its impact on bacterial diversity?  Please discuss and justify.

Thank you for your feedback. To address the impact of antibiotics, we excluded participants who had taken antibiotics in the month prior to the procedure. Additionally, prophylactic antibiotics was not administered for colonoscopy. Further, no participants reported using antibiotics in the one-month post-procedure. Therefore, the impact of antibiotics on our findings is not required.

Please include the figures representing alpha and beta diversity as a whole.

Thank you for this feedback. We have included figures for alpha and beta diversity as supplementary material. We can incorporate into the main text at the editor’s discretion.

What are the conclusions from Figure 1 while it is written in the legend as weak evidence? Please justify.

Thank you for this comment. We have expanded the description and interpretation of the results presented in Figure 1 to enhance clarity and help readers better understand the CLR-transformed abundances. We have also removed our use of the term ‘weak evidence’, and highlighted that these results were statistically significant before, but not after, adjusting for multiple comparisons.

There are many limitations associated with this study. Authors need to include a separate section describing the limitations of the present study.

Thank you for your valuable feedback. In response, we have included a separate section within the discussion specifically dedicated to outlining the limitations of our study. This section aims to clearly address the various limitations associated with our research.

Reviewer 3 Report

Comments and Suggestions for Authors

The title of this article is “Prior appendicectomy and gut microbiota reestablishment in adults after bowel preparation and colonoscopy”. This is an interesting topic, and it is an area that needs our attention. However, there are still some areas of the article that need to be revised:

1. In the discussion, the authors should highlight the possible clinical significance

2. Please.update references in the Introduction and Discussion, the authors should focus on recently published literature (past 2-5 years) and provide a comprehensive and clear comment.

3. The discussion section is lengthy and needs to be streamlined. And the expression needs to be carefully revised.

4. Delete some unnecessary abbreviations (too few occurrences).

5. There are some problems with reference questions, e.g. some journals need to add "..." for abbreviations. Please check.

Comments on the Quality of English Language

Moderate editing of English language required.

Author Response

We would like to thank Reviewer 3 for their feedback which has ultimately improved the quality of our manuscript. Please see below for our responses to their queries:

In the discussion, the authors should highlight the possible clinical significance

Thank you for your feedback. We acknowledge that the clinical significance of our findings remains unclear due to the observational nature and small sample size of our study. In response to your suggestion, we have moved a paragraph from our Introduction to the Discussion section, where we elaborate on the potential clinical significance of appendicectomy. While much more research is needed to determine any definitive clinical significance, our study lays the groundwork for exploring these mechanistic hypotheses. 

Please update references in the Introduction and Discussion, the authors should focus on recently published literature (past 2-5 years) and provide a comprehensive and clear comment.

Thank you for your feedback. We recognise the importance of incorporating recent literature to provide a comprehensive and up-to-date perspective. However, the specific topic we are investigating has limited recent research available. Most foundational and seminal studies in this field were published more than five years ago. While some recent reviews reference these seminal works, highlighting their continued relevance, we have made every effort to include the most pertinent and impactful studies regardless of their publication date. Notably, there have only been three studies specifically exploring appendicectomy and gut microbiota composition, with two published within the past five years (Cai et al. 2021; Sanchez-Alcoholado et al. 2020; Goedert et al. 2014). Additionally, we have updated our discussion to provide a more streamlined and clear commentary on our findings.

The discussion section is lengthy and needs to be streamlined. And the expression needs to be carefully revised.

Thank you for your feedback. In response, we have streamlined and significantly shortened the discussion section. However, we have included a section on the potential clinical relevance of our findings (as requested above), as well as an expanded the discussion on the study’s limitations, as requested by Reviewer 2. Additionally, we have carefully revised the language to ensure that our data are presented as suggestive and preliminary, without overstating the conclusions or relevance of our findings.

Delete some unnecessary abbreviations (too few occurrences).

Thank you for this feedback. We have removed unnecessary abbreviations as requested.

There are some problems with reference questions, e.g. some journals need to add "..." for abbreviations. Please check.

Thank you for highlighting this. We have reviewed the style guidelines and believe that our reference questions are appropriate, however can amend at the editor's discretion.

Round 2

Reviewer 2 Report

Comments and Suggestions for Authors

The authors successfully responded to the reviewer's comments and updated the manuscript accordingly.